# Anatomical Evaluation of Rat and Mouse Simulators for Laboratory Animal Science Courses

**DOI:** 10.3390/ani11123432

**Published:** 2021-12-01

**Authors:** Giuliano M. Corte, Melanie Humpenöder, Marcel Pfützner, Roswitha Merle, Mechthild Wiegard, Katharina Hohlbaum, Ken Richardson, Christa Thöne-Reineke, Johanna Plendl

**Affiliations:** 1Institute of Veterinary Anatomy, Department of Veterinary Medicine, Freie Universität Berlin, 14195 Berlin, Germany; office@myhumanx.com (M.P.); Johanna.Plendl@fu-berlin.de (J.P.); 2Institute of Animal Welfare, Animal Behavior and Laboratory Animal Science, Department of Veterinary Medicine, Freie Universität Berlin, 14163 Berlin, Germany; Melanie.Humpenoeder@fu-berlin.de (M.H.); Mechthild.Wiegard@fu-berlin.de (M.W.); Katharina.Hohlbaum@fu-berlin.de (K.H.); Thoene-Reineke.Christa@fu-berlin.de (C.T.-R.); 3Institute for Veterinary Epidemiology and Biostatistics, Department of Veterinary Medicine, Freie Universität Berlin, 14163 Berlin, Germany; Roswitha.Merle@fu-berlin.de; 4College of Veterinary Medicine, School of Veterinary and Life Sciences, Murdoch University, Murdoch, WA 6150, Australia; K.Richardson@murdoch.edu.au

**Keywords:** evaluation, refinement, education, anatomy, manikin, simulator, laboratory animal course

## Abstract

**Simple Summary:**

Over the past decades simulators of rats and mice have been developed as initial training devices for prospective researchers involved in animal testing. By using these simulators, different techniques such as blood sampling can be be learned prior to working on live animals. As this has the potential to minimize stress and suffering of experimental animals, the use of simulators is demanded by European law. Nevertheless, only little is known about frequency of their use, their anatomical correctness and learning efficiency. With this in mind, a collaborative research project named “SimulRATor” was initiated to systematically evaluate existing rat and mouse simulators. Results will serve as the basis for the development of a new 3D-printed rat simulator with realistic anatomy. In the subproject presented here, simulators were evaluated by experts of the field in order to analyze their anatomical strengths and weaknesses. The evaluation showed, that the limbs and especially the heads were perceived as anatomically unrealistic. Therefore, the authors will focus on these body regions during the construction process by e.g. including movable limbs, and a lower jaw with a tongue. This might positively affect the learning process and outcome and thereby support animal welfare.

**Abstract:**

According to the European Directive 63/2010/EU, education and training involving living rats and mice are classified as an animal experiment and demands the implementation of the 3Rs. Therefore, as a method of refinement, rat and mouse simulators were developed to serve as an initial training device for various techniques, prior to working on living animals. Nevertheless, little is known about the implementation, anatomical correctness, learning efficiency and practical suitability of these simulators. With this in mind, a collaborative research project called “SimulRATor” was initiated to systematically evaluate the existing rat and mouse simulators in a multi-perspective approach. The objective of the study presented here was to identify the anatomical strengths and weaknesses of the available rat and mouse simulators and to determine anatomical requirements for a new anatomically correct rat simulator, specifically adapted to the needs of Laboratory Animal Science (LAS) training courses. Consequently, experts of Veterinary Anatomy and LAS evaluated the anatomy of all currently available rat and mouse simulators. The evaluation showed that compared to the anatomy of living rats and mice, the tails were perceived as the most anatomically realistic body part, followed by the general exterior and the limbs. The heads were rated as the least favored body part.

## 1. Introduction

In the scope of the 3R principle (Refine, Reduce, Replace) by Russell and Burch, animal experiments must be replaced by alternatives whenever possible. However, if an animal experiment cannot be replaced, the number of animals must be reduced and procedures must be refined, in order to minimize the animals’ pain, suffering, and distress caused by the experiment [1]. Since education and training involving live animals, e.g., Laboratory Animal Science (LAS) courses, are regarded as animal experiments, it is compulsory to implement the 3R principle. In LAS courses, prospective experimenters acquire the necessary knowledge and practical skills for the conduction of experimental procedures on animals. Participants are taught the basic theoretical knowledge and practical techniques, such as animal handling, sample collection, substance administration, anaesthesia, and euthanasia. When these procedures involve living animals, they are classified as animal experiments. This is because distress and pain cannot be fully excluded, especially as course participants are often untrained and inexperienced [2]. Sometimes participants come from disciplines that do not involve animal handling. Such unpreparedness can be exacerbated by family and cultural mindsets, as fear of certain animal species may predispose course participants to anxiety before and during training procedures [3,4].

As rats and mice are the most frequently used species in animal experiments in the European Union, they are also commonly used for educational and training purposes. According to the Summary Report on the statistics on the use of animals for scientific purposes in the Member States of the European Union and Norway from 2018, a total of 166,437 animals including 84,059 mice and 41,216 rats were used for educational purposes in Europe [5]. In Germany, 53,805 animals (35,557 mice and 9589 rats) were used for “higher education or training for the acquisition, maintenance or improvement of vocational skills” in 2019 [6].

According to the Directive 2010/63/EU, the aforementioned situation constitutes an urgent need for the application of the 3Rs in the field of experimental education and training [7]. The implementation of simulator training has the potential to improve the present situation. Simulators are intended to closely approximate real-life situations, enable active learning and offer a safe environment for repeated practice and error. Consequently, simulator users can make mistakes or fail a procedure and practice until they master it without causing additional burden to animals [8,9].

Over the past two decades, human medical education has witnessed an increasing use of simulator technology. Without the involvement of patients, they are readily available at any time, capable of reproducing specific exercises on demand, e.g., different procedures, as well as clinical conditions, and thereby provide consistent and specific hands-on training [10]. Although the field of simulator training in veterinary medicine is not as advanced as in human medicine [11], veterinary simulators are becoming more common, e.g., for bovine and equine rectal palpation [2,12], feline abdominal palpation [13] and canine laparoscopic training [14]. Practicing manipulations on simulators can potentially minimize stress, pain, and suffering in animals used in education, because those who train the techniques are better prepared prior to training on animals. As the simulators serve as substitutes for living animals, their anatomy should be as realistic as possible, regarding both visual appearance and tactile perception [15]. Besides the above-mentioned refinement aspects, simulators have the potential to reduce the number of laboratory animals used in training and education. For LAS, only a few rat and mouse simulators are commercially available for practical LAS training to date, developed and produced in the United Kingdom, the United States, and Japan. These include six rat simulators and a single mouse simulator. One rat simulator, specifically designed for microsurgical training, was excluded from this evaluation, as it cannot be utilized to train other handling and procedural techniques [16].

Little is known by now about the frequency of use, anatomical correctness, learning efficiency, and practical suitability of these simulators. With this in mind, a collaborative research project named “SimulRATor” was initiated, in which a team of anatomists, laboratory animal scientists, an epidemiologist and a medical engineer were assembled to systematically evaluate existing rat and mouse simulators. The evaluation consisted of three subprojects: (1) A user-oriented online survey for course trainers, to determine the level of awareness as well as implementation, satisfaction and requirements for simulators used in LAS courses [17]; (2) An anatomical evaluation of currently available simulators for rat and mouse and; (3) A tutored simulator training with participants of LAS courses, in which their learning success with the currently available simulators and their demands and requirements for future simulators was assessed. The results of the three subprojects will eventually be analyzed and serve as the basis for the construction development of a new costeffective 3D-printed rat simulator with realistic anatomy and haptics (the perception of objects by touch and proprioception). The new rat simulator is intended as a training model for handling, restraint and procedural techniques, such as blood sampling, oral administration or subcutaneous injection but not for interventions on a surgical level.

The objective of subproject 2 presented here was to identify the anatomical strengths and weaknesses of the available rat and mouse simulators and to determine anatomical requirements. Consequently, experts of Veterinary Anatomy and LAS evaluated all currently available rat and mouse simulators anatomically and according to procedural specifications using specifically designed questionnaires. In these, the simulator’s general exterior, head, tail, and limbs were compared with the anatomy of living rats and mice.

## 2. Materials and Methods

### 2.1. Simulators

At the beginning of the project in 2018, a market analysis was conducted using generic internet search engines and databases for alternative learning such as the Norwegian Inventory of Alternatives (NORINA) [18] or the International Network for Humane Education (InterNICHE) [19]. Therefore, the authors could identify the commercially available rat and mouse simulators (Table 1). For the project, two examples of each simulator were purchased, including spare parts such as replacement tails and artificial blood. The following specifications of the simulators are based on information provided by the manufacturers. Rat simulators A and B and the mouse simulators were produced in Japan. Rat simulators C and D originated from the United States and Rat simulator E was from the UK.

Although this paper focuses on identifying anatomical requirements for a new rat simulator, the mouse simulator was included as the results may be relevant for a novel mouse simulator in the future. Nevertheless, the evaluation results of the mouse simulator were not directly compared to the results of the rat simulators.

#### 2.1.1. Rat Simulators

##### Rat Simulator A

Rat simulator A (Figure 1) is a white silicon rat. It consists of a head, torso with four limbs, and a tail. It possesses a dorsal skin fold for scruffing. The limbs allow the simulator to stand in an upright posture. The head has an oral cavity with two upper and two lower incisors, a tongue, and a large oval-shaped opening to allow training of oral administration. To determine whether the feeding tube for oral administration is correctly placed into the stomach or incorrectly inserts into the trachea, a transparent plastic section is imbedded in the abdominal wall, through which a transparent stomach and trachea can be seen. When correctly administered into the stomach, the infused fluid leaves the rat model through a small opening in the inguinal region. The detachable white tail has two lateral vessels for intravenous application and blood sampling using imitation blood.

##### Rat Simulator B

Rat simulator B (Figure 2) is a beige-colored rat simulator made of silicon and soft vinyl chloride, which according to the manufacturer mimics a 9 month-old male Sprague Dawley Rat. From a functional point of view, it seems to be very similar to rat simulator A. In contrast to rat simulator A, it differs in body morphology. Its posture is curved, its skin is harder, and it has no teeth. Regarding the functionalities, no obvious differences compared to rat simulator A exist.

##### Rat Simulator C

Rat simulator C (Figure 3) is a white silicon rat model designed primarily for endotracheal intubation. It consists of a body with four limbs and a non-removable pink tail having one central blood vessel that allows the collection and administration of imitation blood. Its wide-open oral cavity houses two upper incisors and a tongue; posteriorly, there is a circular opening. This simulator has robust nylon whiskers around the nose and plastic claws on the limbs. It is the only simulator that offers the opportunity of blood sampling from the heart and the saphenous vein.

##### Rat Simulator D

Rat simulator D (Figure 4) is a brown model lined with fur. It has a solid body with four limbs and a detachable pink tail with one central blood vessel for intravenous application and blood sampling. In addition, it has silicon feet, robust nylon whiskers, and glass eyes painted to indicate their pupil and red iris. There is no oral cavity. According to the manufacturer, the simulator is made for common handling and restraint techniques. In addition, its replaceable ears can be used to train ear tagging.

##### Rat Simulator E

Rat simulator E (Figure 5) is a white silicon rat simulator with four flexible limbs, independent toes and a detachable tail with two lateral blood vessels for intravenous application and blood sampling. The head has solid red eyeballs. In addition, it is equipped with a movable spinal column and flexible skin for various techniques, including handling, restraint, subcutaneous and intramuscular administration as well as microchipping. Its oral cavity houses upper and lower incisor teeth, a tongue and a hard palate and provides training of oral administration. In addition, it has an anal opening for temperature measurements using a thermometer.

#### 2.1.2. Mouse Simulator

To the best of our knowledge, the mouse simulator (Figure 6) currently is the only commercially available mouse simulator. It has flexible skin to allow training of handling and restraint techniques. The tongue- and toothless oral cavity provides the possibility to train oral administration. There is a tail with two lateral vessels, allowing intravenous administration. The toes are made of solid plastic and those on the forepaw have a curved claw, which can hold on to the cage grid.

### 2.2. Evaluators and Questionnaires

The group of evaluators consisted of 14 veterinarians specialized in veterinary anatomy (*n* = 10) or laboratory animal science (*n* = 4), mainly from the Berlin area who participated voluntarily. They evaluated the available simulators independently throughout the evaluation process. In the beginning, each evaluator was briefed on the procedure and the questionnaires. For every simulator, two different paper-based questionnaires in German language had to be completed without a specific time limit. Questionnaire A (Appendix A Appendix A) consisted of four 6-point Likert items and three open text questions. On the 6-point Likert scale [20], evaluators were asked to rate how realistic they considered the anatomical features of different body parts in comparison to a living rat and, in the case of the mouse simulator, to a living mouse, ranging from “very realistic” (1) to “very unrealistic” (6) (Appendix A Appendix A). If a certain anatomical feature was not present in a simulator, it was rated as “not applicable” (7). The body regions included the general exterior, e.g., asking about the body size and proportions, whilst the following parts were specifically related to the head, tail and limb region. Questionnaire A included the three following open questions:From an anatomical point of view, what did you particularly like about rat simulator X?From an anatomical point of view, what did you particularly dislike about rat simulator X?What would you improve from an anatomical point of view in rat simulator X?

Questionnaire B (Appendix A Appendix A) consisted of a table, in which the evaluators could compare the five rat simulators with each other and rank them based on their overall anatomical appearance from “most realistic” to “least realistic” (Appendix A Appendix A). In order to prevent mistakes, every simulator was assigned a number that also had to be indicated within the ranking table. The mouse simulator was not included in the ranking as a direct comparison between mouse and rat simulators seemed inappropriate.

### 2.3. Statistics

The descriptive results are either depicted in diverging stacked bar charts, showing the frequencies of the evaluators’ answers, or presented in tables as mean values over all given answers. The answers of the lowest and highest rated simulator regarding the four body parts are presented in diverging stacked bar charts, which were created using Microsoft Excel (Windows Corporation, 2018 version, Redmond, WA, USA). The values in the tables are the arithmetic mean of all given ordinal-scaled variables. The overall mean value itself is the arithmetic mean of all given mean values of all parameters. Using IBM SPSS Version 25 (IBM Corporation, Armonk, NY, USA), a one-way analysis of variance (ANOVA) was calculated (Appendix A Appendix A) to determine whether the evaluation results of the rat simulators differed significantly. The Dunnett post hoc test was used with rat simulator A being the reference group. Model diagnostics included visual inspection of normality and homoscedasticity of residuals. The limb scores were not normally distributed. Thus, a Kruskal-Wallis test with the respective Bonferroni adjustment was used. A *p*-value lower than 0.05 was considered to indicate significance. The mean ranks, corresponding the evaluation results of questionnaire B, are given as an arithmetic mean.

## 3. Results

### 3.1. Rat Simulator A

The general exterior of rat simulator A received a mean score of 3.35, which represented the lowest and therefore best mean score according to closeness to reality of all simulators (Appendix A Appendix A and Table 2). Body size, weight, shape, and proportions were rated best among all rat simulators. The head (Figure 7), with a score of 3.61, ranked 2nd among all simulators (Appendix A Appendix A and Table 3). Here, the head shape and proportions were considered to be “quite realistic”. The mobility of the lower jaw and the ears was perceived as “rather unrealistic”. With a mean score of 2.84, the tail (Figure 8) was evaluated the 2nd best among all simulators (Table 4). The blood vessels of the tail received “rather” and “quite realistic” scores. Like the tail, the limbs (Figure 9) were also rated 2nd best (Table 5). Their proportions and posture were considered to be “rather realistic”, length and toes “rather unrealistic”.

Concerning the first open question on what was liked about rat simulator A, the most frequent answers were: tail, haptic and proportions (5/14). It was stated that “… the tail reflects the actual anatomy of a live rat relatively well” and that rat simulator A possesses an “… overall quite natural body form and posture”.

In the second open question concerning what was particularly not liked about rat simulator A, the ears were the most frequently mentioned body parts (5/14), as they were perceived as “… too thick” and that rat simulator A additionally has a “… lack of detail on the limbs”. Furthermore, the oral cavity was regarded as an unrealistic body region.

In accordance with the presented answers, evaluators recommended improving the limbs (7/14), especially the mobility and posture, the oral cavity and mouth opening (7/14), the ears (6/14), and the toes (5/14) in rat simulator A. One evaluator suggested supplementing rat simulator A with the following details of rat simulator E: toes, joints, mouth, ears, skin and to “… remodel the mouth opening to be more flexible”. Another evaluator proposed a similar approach by stating that the tail of rat simulator B and the body of rat simulator A should be combined into one simulator.

In questionnaire B, in which the evaluators should rank the rat simulators according to their overall anatomical appearance, rat simulator A had a mean rank of 1.36 and was therefore considered the most realistic among all rat simulators included in the evaluation (Table 6). Ten of 14 evaluators ranked rat simulator A at 1st place.

### 3.2. Rat Simulator B

With a mean score of 3.67, the general exterior of rat simulator B ranked 2nd among all rat simulators. Although the body size and proportions were evaluated favorably, its haptic, mobility and consistency of the skin and the mobility of the joints were perceived as “rather unrealistic”. The head ranked 3rd among all simulators, receiving a mean score of 4.83. In particular, the mobility of the head, the mouth and pharyngeal–laryngeal area including the tongue and the mobility of the lower jaw were considered as “quite” or even “very unrealistic”.

With a mean score of 2.78, the tail of rat simulator B was rated best of all simulators (Appendix A Appendix A and Table 4). Here, especially the size, position, and course of the blood vessels were regarded as “rather realistic”. With a mean score of 4.49, the limbs ranked 4th. All parameters were at least considered “rather unrealistic”, while the haptic and mobility were valued as “quite unrealistic” and the toes of the limb as “very unrealistic”.

In the answers to the open questions, the tail and its vessels were highlighted most positively (9/14), whereas the skin (8/14) was seen as negative. This was also true for the limbs and toes (8/14) stating that they lack realistic consistency and posture. Moreover, it was commented, that the “… mouth opening is too far open to be realistic”. In addition, the ears were considered to be “… too small and thick to be realistic”.

Asked what they would change in particular in rat simulator B, six of 14 evaluators referred to the limbs and the body posture (5/14) by saying that the “… rat should stand in a natural posture”. The evaluators suggested the inclusion of a more realistic oral cavity with teeth (6/14).

In questionnaire B, rat simulator B scored a mean rank of 2.29 and was considered the 2nd most realistic rat simulator in overall anatomic appearance.

### 3.3. Rat Simulator C

With a mean score of 5.21, rat simulator C ranked 5th concerning the general exterior (Appendix A Appendix A and Table 2). All parameters were considered “rather unrealistic”, the haptic and skin mobility “very unrealistic”. The head region ranked 4th among all simulators. The mean score of the tail was 4.68, as all parameters were considered either “rather” or “quite unrealistic”. The connection of the tail to the torso as well as the consistency and the size of the tail blood vessels received the least favorable ratings. With a mean score of 5.36, the limbs with their overall appearance, haptic, and posture were rated the least anatomically correct among all rat simulators (Appendix A Appendix A and Table 5).

In the open questions, the presence of whiskers was positively indicated (3/14) whereas the overall appearance (5/14), the head (4/15), and the haptic as well as proportion of the limbs (4/14) were mentioned negatively.

Answers concerning potential improvements of anatomy were heterogeneous, amongst others included the limbs (5/14) and the position of the blood vessels (4/14) were listed. It was commented to “… add joints and use a firmer material for the limbs to limit its mobility”.

In the ranking of the general anatomical appearance, rat simulator C had a mean rank of 4.43, sharing 4th place with rat simulator D.

### 3.4. Rat Simulator D

In questionnaire A, with a mean score of 4.96, the general exterior of rat simulator D ranked at 4th place. Here, the position, course and consistency of the blood vessels were perceived as “very unrealistic”, whereas body size was regarded as “rather realistic”. The head, with a score of 5.60, ranked last among all rat simulators, however, the ears were perceived as being “rather realistic”. The absence of an oral opening and cavity was considered as a disadvantage. With a mean score of 4.85, the tail ranked 5th (Appendix A Appendix A and Table 4). The limbs ranked 3rd among all simulators receiving “rather” and “quite unrealistic” scores.

In the open question, the evaluators mentioned the ears positively (5/12).

Whereas they commented critically, that the inner structure of rat simulator D is made out of hard and stiff material (7/14) and that the connection of the tail and trunk does not look realistic. Likewise, there were negative comments on the limbs (5/14) having no proper connection to the trunk and being only attached by skin and fur.

The evaluators most often suggested improving the haptics (9/14) and skin (7/14) by using “more realistic fur or no fur at all”. Moreover, the limbs (6/14) “… should have their origin more cranially located”.

In questionnaire B, rat simulator D achieved a mean rank of 4.43 and therefore shared the equal ranking place with rat simulator C.

### 3.5. Rat Simulator E

With a mean score of 3.99, the general exterior of rat simulator E ranked in 3rd place. The body size and mobility of the skin were rated and considered “rather realistic”. The consistency of the blood vessels and the mobility of the joints were the least favorable rated parameters. The mobility of the limb joints was regarded as “quite unrealistic”. In contrast, the head of rat simulator E was rated the most realistic among all simulators with a mean score of 3.51. In particular, the overall appearance of the head, the head shape and proportions were considered as “rather realistic”. This also applied to the mobility of the lower jaw, which was perceived as the best among all simulators. The ears were the best-rated parameter of the head. With a mean score of 3.08, the tail of rat simulator E was considered the 3rd most realistic among all simulators. Here, especially the haptic, mobility and length of the tail received the best ratings and were considered “quite realistic”. Although all parameters were at least considered “rather unrealistic”, the limbs of rat simulator E ranked 1st among all simulators (Appendix A Appendix A and Table 5). The toes of the limbs received the best rating among all rat simulators.

In the open questions, the evaluators explicitly singled out the quality of the tail and its vessels (8/14), the movable spinal column (5/14), the ears and the movable lower jaw (both 4/14).

In contrast, the body proportions and posture (6/14) were perceived negatively.

Asked what they would change in particular in the simulator, five of 14 evaluators said that they would implement both an opening for the trachea and the oesophagus in the oral cavity (5/14). Furthermore, body proportions and the lack of mobility of the limbs were suggested features to be improved.

In questionnaire B, rat simulator E had a mean rank of 2.50 and therefore was considered overall the 3rd most realistic rat simulator. Four of 14 evaluators ranked rat simulator E at 1st place, seven out of 14 voted it at the 3rd place.

### 3.6. Mouse Simulator

The general exterior of the mouse simulator achieved a mean score of 3.63 The body size and weight were considered “quite realistic” followed by the mobility of the skin. The worst-rated parameter by far was the mobility of the limbs. Most parameters regarding the head were rated as “rather unrealistic” or even less favorable. The oral cavity with the appearance and degree of the mouth opening was rated as “quite unrealistic”. The eyes, the mobility of the lower jaw and the pharynx-larynx region were rated as “very unrealistic”. Overall, the head of the mouse simulator had a mean score of 5.12. The tail, besides the consistency of the blood vessels, was overall rated “rather unrealistic” and “quite unrealistic”. Here, the length of the tail was considered the best parameter. The limbs received a mean score of 4.54. Especially the overall appearance, the haptic and the mobility of the limbs were assessed negatively.

In the open question, the mobility of the skin (8/14), the overall haptic (6/14) and the tail (5/14) were mentioned positively.

In contrast, the limbs (6/14) were perceived as too short with too much focus on the paws. In addition, it was commented upon negatively that, the limbs “… are not separate but part of the trunk”. Five of 14 evaluators mentioned that they particularly did not like the oral cavity.

Improvements proposed in the open questions included the “… shape the opening of the oral cavity” (6/14), the mobility of the head (6/14), and the implementation of a trachea and an oesophagus (6/14).

In the ranking question of questionnaire B, only rat simulators were included.

### 3.7. Analysis of Variance

ANOVA models were developed separately for the general exterior, head, tail, and limb. The mean ratings were compared between the animal models with rat A being the reference group (Appendix A Appendix A). Comparing the general exterior, tail scores and head scores, rat simulator A had significantly lower values than rat simulator C and rat simulator D but not from B and E. The limb scores were not normally distributed. A Kruskal–Wallis test revealed significantly higher values only for rat simulator D.

## 4. Discussion

The objective of this study was to evaluate the anatomical correctness of rodent simulators used for LAS courses in order to determine anatomical requirements necessary to develop a new anatomically correct rat simulator for handling and basic procedural techniques. Therefore, the discussion focuses on the evaluation results of the rat simulators and the requirements and recommendations derived from them.

The authors are aware that the sample size of evaluators is rather small, but it has to be considered that the number of available veterinarians specialized in anatomy is limited to the Berlin area.

The evaluators generally found that the rat simulators were characterized by an unrealistic anatomy of their general exterior (overall mean score = 4.24). As such, none of the simulators evaluated would be appropriate as a base model, but instead, the anatomy of a real rat should remain the gold standard. As a virtual 3D model is required for the 3D printing process of the novel rat simulator, a micro-CT (micro-computed tomography) dataset of a middle-aged rat could serve as an ideal template to assure anatomical correctness.

In the open questions, several evaluators commented that it would be desirable to achieve realistic haptics in a novel simulator, a combination of both solid materials for inner structures such as bone, and softer material for outer structures such as musculature and skin. This can be achieved by using the newly evolving technology of Multi-Material 3D printing, where both soft and hard materials can be processed simultaneously [21]. As the mobility of the simulators was criticized in the evaluation, the feasibility of introducing atlanto–occipital and atlanto–axial joints, important for pitch motion and rotation of the head, or of an entire vertebral column should be considered. All simulators were missing a rib cage, although its inclusion could be beneficial, as one can constantly feel it during handling techniques such as the “over-the-shoulder grip”. Moreover, the rib cage is important for cardiac puncture [22], which can be performed from various angles, e.g., through the intercostal space or from underneath the costal arch. The xyphoid process (the most caudal point of the sternum) is important when estimating the ideal length of the feeding tube for oral administration [23].

In the evaluations, the head was the least favorable body part, with a mean score of 4.54. From an anatomical point of view, this needs to be improved dramatically in a novel simulator. Despite reasonable shape and proportions of the head, the evaluators perceived the mobility of the lower jaw and the appearance of the oral cavity as unrealistic. This is problematic, as an anatomically erroneously structured oral cavity would negatively affect the learning processes of challenging techniques, such as oral administration or endotracheal intubation. In addition, these techniques are potentially associated with adverse consequences, such as oesophageal trauma and aspiration pneumonia [24,25]. Some simulators in this study had wide-open oral openings, which falsely facilitates the insertion of feeding tubes. Consequently, a movable lower jaw that is associated with a realistically opening and closing of the mouth is important. The anatomical appearance would thus be improved by the presence of a flexible tongue as well as upper and lower incisors, comparable in size to those of living rats. A larynx with an epiglottis-like entry should lead to a trachea characterized by semi-solid rings, mimicking normal tracheal cartilages of a live rat. Dorsal to the trachea, a separate oesophagus, with a realistic diameter needs to be implemented.

Several studies have demonstrated the importance of correct anatomy of the upper airway in human simulators. Yang et al. compared four human simulators for laryngoscopic orotracheal intubation with fresh frozen human cadavers and reported better outcomes from training undertaken on cadavers [26]. Schebesta et al. assessed how anatomically realistic the upper airways of six different human patient simulators were. They found major differences in comparison to an actual patient’s anatomy and concluded that especially for inexperienced personnel, an unrealistic airway may lead them to acquire inappropriate airway management techniques. These circumstances can negatively affect the training outcome, as falsely trained procedures are very difficult to unlearn and failures in managing an airway are associated with a high risk of mortality. This of course also questions the appropriateness of translating simulation-based research and training into the care of real patients [27,28]. Working with living animals is much more challenging compared to human patients, as animals often show abrupt movements and sometimes aggression, which additionally poses a risk for bite injuries.

The participants in this study found that the tail of the simulators was the most anatomically realistic of all evaluated body parts (overall mean score = 3.65). The correct anatomy of the simulator’s tail such as its length, mobility, skin texture, as well as the tail vein position and structure are important in learning routine techniques, including venepuncture for blood sampling and drug administration [29]. However, it was only considered “rather realistic” at best, while the tails of rat simulators C and D were perceived as “rather unrealistic”. This suggests that there is still room for improvement of the tail of a novel rat simulator.

In the open questions, the limbs (overall mean score = 4.47) were one of the most criticized body parts. The proportions and posture of the limbs of rat simulator A were the only parameters of all simulators that were rated as “rather realistic”. While all limbs were overall rated as “rather unrealistic”, the best mean score was that of rat simulator E. The manufacturers of both simulators used quite different approaches. The semi-hard but rather immobile limbs of rat simulator A would be disadvantageous in the training of restraint techniques such as the “under-the-shoulder grip”, but allows the simulator to stand in an upright and stable position. The limbs of rat simulator E are significantly firmer but more unstable, which makes positioning the simulator in an upright posture difficult. Nevertheless, their better limb mobility allows a closer approximation to the body, which is required for some restraint techniques. The novel simulator should therefore combine the beneficial limb characteristics of both simulators. It should be able to stand in an upright and stable position, but at the same time maintain the mobility of its limbs. As the feasibility of a bone-like skeleton should be addressed in a novel simulator, a focus on the inclusion of limb joints should also be considered, as the limbs should be as mobile as possible.

To our knowledge, no anatomical evaluation of veterinary simulators has been published until today. Nevertheless, false training on veterinary simulators could negatively affect animal well-being by causing complications when initially performing the techniques on live animals, which contradicts the aims of the 3Rs. Especially for rats and mice, due to their small body size, severe adverse effects could be caused when needles or feeding tubes are placed incorrectly, resulting in distress and pain and might jeopardize the life of the animal [30]. Moreover, insufficient or even false training might prolong the duration of procedures trained on live animals and expose them to even higher stress and pain levels.

The authors are aware that there is a thin line between anatomical correctness and functionality and that occasionally some anatomical correctness may need to be neglected to maintain or improve functionality. Therefore, one could argue, that it may not be necessary that the entire anatomy of the simulator is developed in great detail, but only the areas relevant for training, e.g., the oral cavity. Walshaw (2004) corroborated this by stating that inexpensive models can be used in LAS courses to teach humane handling techniques and methods of holding and using instrumentation or equipment. Here, she suggests that restraint of rodents can be performed easily using a sock filled with a soft material such as cotton. Pieces of fruit such as oranges can be used to practice injection techniques and how to carefully manipulate the syringe and needle and thereby develop hand–eye coordination skills [3]. On the other hand, simulators for mice and rats have a small body size and during the training, the user is in contact with every part of the animals’ body. Therefore, it is even more important that all body regions are anatomically correct.

Simulator fidelity when considering biological modelling is defined as the extent to which the appearance and behavior of the simulator match the appearance and behavior of the simulated system. The fidelity of simulators ranges from low-fidelity models, which include static manikins, to high-fidelity simulators with life-like manikins connected to controlling computer systems [31,32,33]. Therefore, low-fidelity simulators are distinguished from high-fidelity simulators mainly by their functionality and the possibility of controlling the success of the related training or procedure [34]. Higgins et al. pointed out that “… it is pointless to build a training simulator that doesn’t provide useful feedback on performance to the trainee” [35]. On the other hand, a study comparing low- and high-fidelity simulators with each other could not detect significant differences between both simulator systems [36]. The implementation of blood vessels and the use of artificial blood can be regarded as an advanced learning success indicator, as correct blood sampling enables the withdrawal of the blood and provides a more visually realistic training experience. In addition, the transparent abdominal panel of rat simulator A and B, showing the trachea and oesophagus, can be regarded as success control. Therefore, it can be stated that a functional and thought-out simulator must possess several indicators for success control. Whether the authors will use mechanically or electronically implemented systems or a combination of both in the novel simulator, e.g., in form of a microcontroller, is not decided yet.

The overarching question of whether simulator-based training really has a positive learning effect is not easy to answer in general. Although many evaluations in various fields were carried out, the literature shows conflicting results. In some cases, a positive training effect could be determined [37,38], while other studies showed that it is not always beneficial [31,39,40].

As simulators for rats and mice are highly specialized products, relevant research and development is scarce. It is, however, necessary for evaluation and further assessment to be thoroughly conducted and published in order to analyze efficiency and implementation and to promote these alternative training methods. An important fundamental task for further refinement in education and training is the development of prototypes, e.g., simulators, which can have lasting effects on the 3Rs. In order to prepare the user for animal experiments in an effective way, it is crucial that simulators display correct and realistic anatomy. For the novel simulator, all efforts and resources will be exhausted to anatomically mimic a live rat as realistically as possible in order to optimize training outcome and therefore animal welfare.

## 5. Conclusions

The evaluation showed that the heads were the least anatomically realistic body parts of the rat simulators. Therefore, the authors are convinced that a movable lower jaw, a realistic mouth opening, a flexible tongue as well as upper and lower incisors must be implemented in the new simulator. Thus, including a larynx with an epiglottis, a trachea with semi-solid rings, mimicking normal tracheal and a separate oesophagus, might hold the potential to positively affect the learning processes and outcome of procedural techniques such as oral administration or endotracheal intubation. By using the Multi-Material 3D printing technology for the construction, solid materials for inner structures such as head, neck and limb joints, a vertebral column and a rib cage, and softer material for outer structures such as musculature and skin can be processed simultaneously. Therefore, the overall haptic and movability of the novel simulator could benefit and the training of handling techniques might improve significantly. The authors moreover aim to include several indicator systems to enable useful feedback and quantified learning success for trainees and trainers.

## Figures and Tables

**Figure 1 animals-11-03432-f001:**
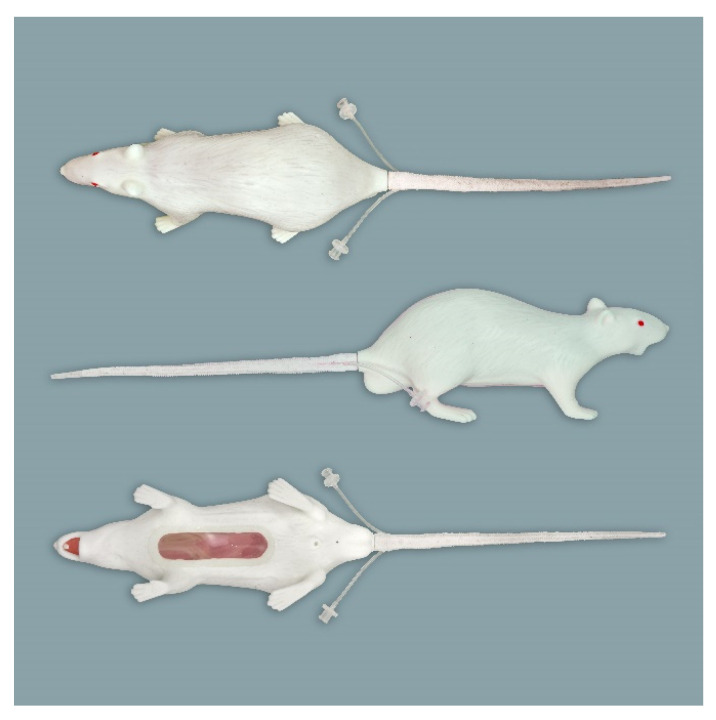
Rat simulator A in dorsal, lateral, and ventral view.

**Figure 2 animals-11-03432-f002:**
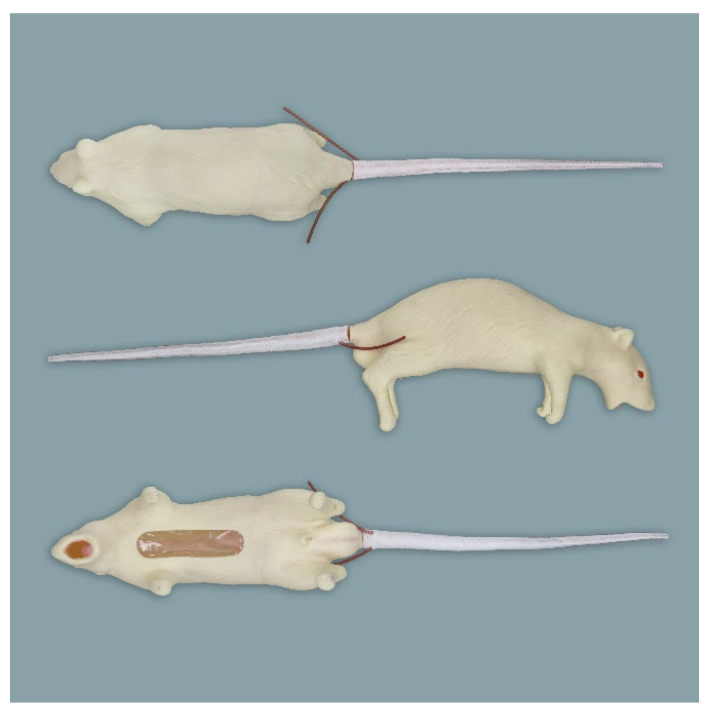
Rat simulator B in dorsal, lateral and ventral view.

**Figure 3 animals-11-03432-f003:**
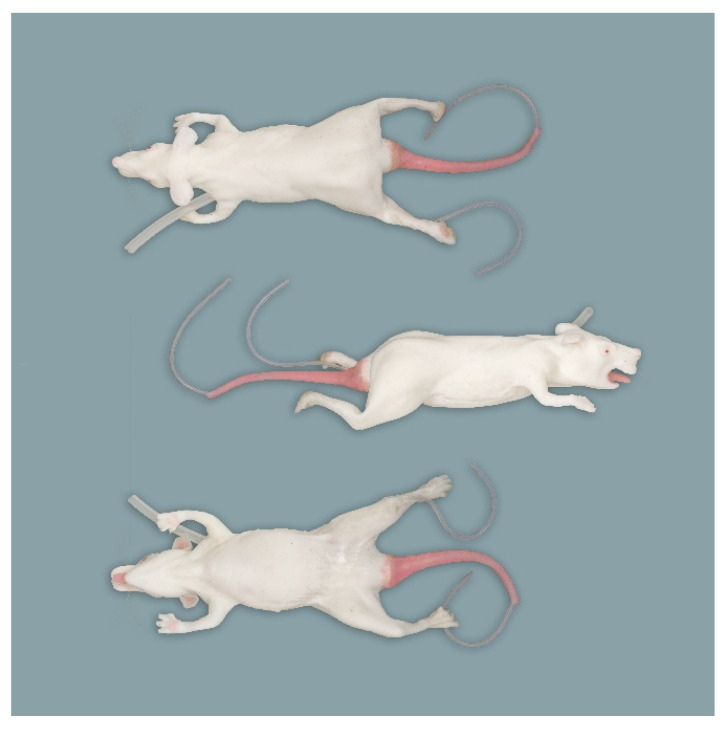
Rat simulator C in dorsal, lateral and ventral view.

**Figure 4 animals-11-03432-f004:**
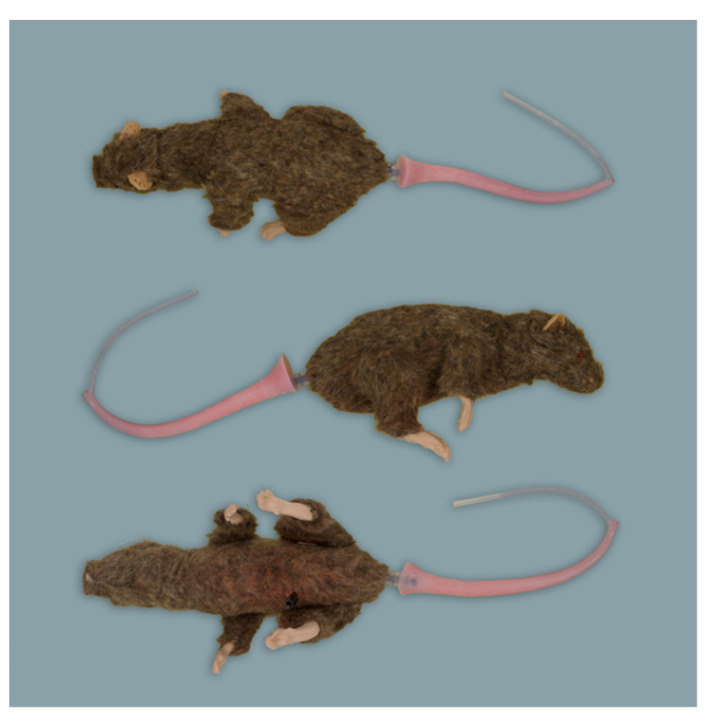
Rat simulator D in dorsal, lateral and ventral view.

**Figure 5 animals-11-03432-f005:**
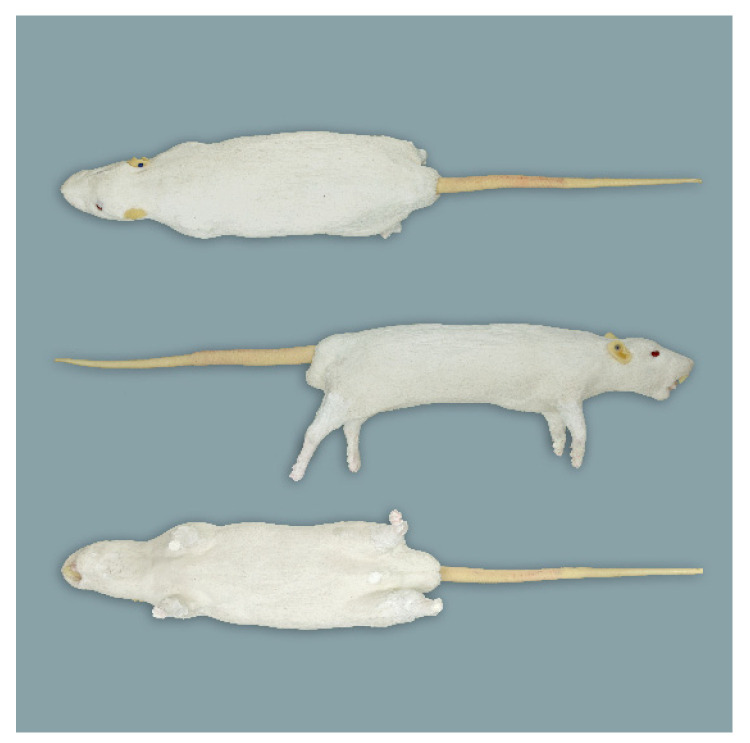
Rat simulator E in dorsal, lateral and ventral view.

**Figure 6 animals-11-03432-f006:**
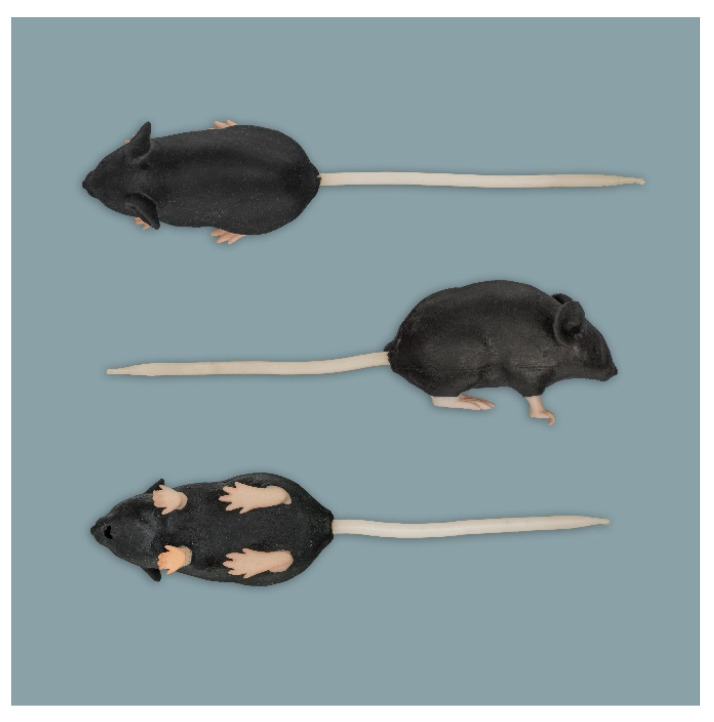
Mouse simulator in dorsal, lateral and ventral view.

**Figure 7 animals-11-03432-f007:**
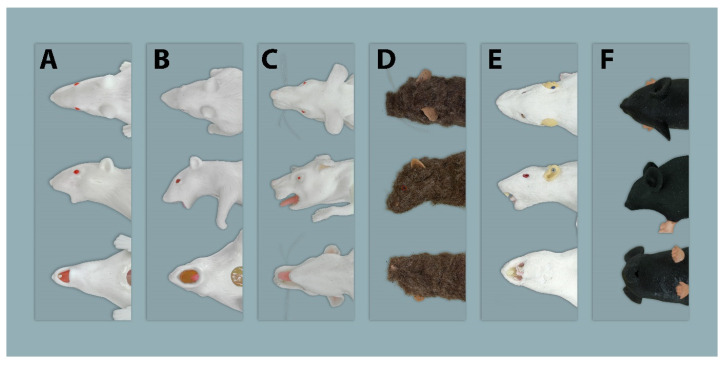
Dorsal, lateral, and ventral view of the heads of all evaluated simulators. (**A**–**E**) = Rat simulators; (**F**) = Mouse simulator.

**Figure 8 animals-11-03432-f008:**
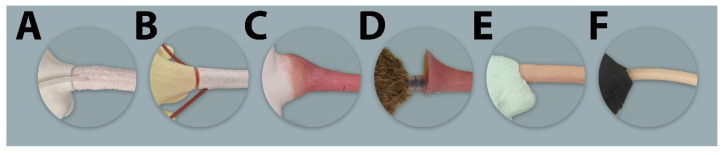
Lateral view of the tails’ connection to the trunk of all evaluated simulators. (**A**–**E**) = Rat simulators; (**F**) = Mouse simulator.

**Figure 9 animals-11-03432-f009:**
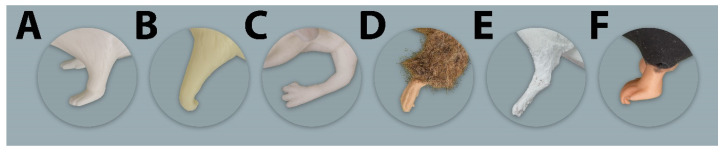
Lateral view of the forelimbs of all evaluated simulators. (**A**–**E**) = Rat simulator A–E; (**F**) = Mouse simulator.

**Table 1 animals-11-03432-t001:** Overview of the rat and mouse simulators for handling and basic techniques evaluated in this study. Product names of the simulators were anonymized.

Simulators	Rat Simulator A	Rat Simulator B	Rat Simulator C	Rat Simulator D	Rat Simulator E	Mouse Simulator
General exterior	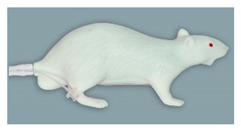	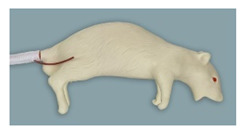	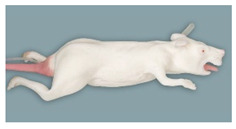	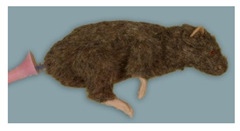	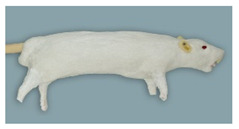	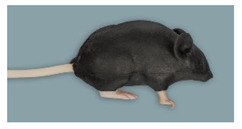
Body weight [gram] ^1^	220	150	550	410	350	35
Body length, width ^2^, height ^2,3^ (L × W × H) [cm]	16.5 × 4.5 × 6.5	17.0 × 4.5 × 7.0	21.5 × 6.0 × 5.0	21.0 × 4.5 × 6.5	25.0 × 6.5 × 8.0	7.5 × 2.5 × 3.0
Tail length [cm]	18	18.5	16.0	16.5	17.5	8.5
Limb length ^4^ (forelimb /hindlimb) [cm]	3.5/2.5	4.0/5.0	6.5/10.5	6.5/9.5	5.0/7.0	1.5/1.0
Materials	silicon	silicon; soft vinyl chloride	silicon	hard plastic (inner body and eyes); fur-like material	hard plastic (head and eyes); foam (inner body); silicon (outer body layer); plastic wire (inner limb structures)	silicon; hard plastic (paws)
Training options	A; B; E; F; I	A; B; E; F; I	A; B; C; D; F; I	A; B; F; J	A; B; E; F; G; H; K; L	A; E; F

^1^ excluding tail; ^2^ at the level of the lower abdomen; ³ when positioned in the natural posture of the simulator; ^4^ measured as vertical line from the most dorsal part (protruding from the body) to the toes. A = handling and restraint; B = blood sampling via tail vein; C = blood sampling via saphenous vein; D = blood sampling from the heart; E = administration by oral gavage; F = intravenous administration via tail vein; G = subcutaneous administration; H = intramuscular administration; I = endotracheal intubation;.J = ear punch; K = micro chipping; L = temperature measurement.

**Table 2 animals-11-03432-t002:** Anatomic parameters with arithmetic mean values based on the ordinal-scaled variables of the general exterior. Value ranges are as follows: 1–1.49 = very realistic (dark blue; not depicted); 1.5–2.49 = quite realistic (gray blue); 2.5–3.49 = rather realistic (light blue); 3.5–4.49 = rather unrealistic (light orange); 4.5–5.49 = quite unrealistic (red); 5.5–6.5 = very unrealistic (dark red); 7 = not applicable (gray).

General Exterior	Simulators					
Simulator	Rat A	Rat B	Rat C	Rat D	Rat E	Mouse
Parameter						
Overall appearance	2.93	3.21	5.21	4.93	3.64	3.14
Haptic	3.57	4.50	5.57	4.71	4.00	3.07
Mobility of the skin on the neck	3.79	4.29	5.93	4.36	3.14	2.57
Mobility of the skin on the flank	3.86	4.43	6.07	5.00	3.71	2.57
Consistency of the skin surface	4.07	4.93	5.36	4.50	4.00	3.29
Body size	2.14	2.71	3.85	2.86	3.21	2.43
Body weight	2.21	3.29	4.29	3.86	3.57	2.29
Body shape	2.07	2.93	4.93	4.57	3.86	2.71
Proportions	2.14	2.79	4.64	4.71	3.71	3.00
Gender-specific characteristics	3.21	3.29	7.00	7.00	3.92	7.00
Blood vessels	3.86	3.71	4.57	5.62	4.62	4.08
Position of the blood vessels	3.29	2.86	4.64	5.62	4.07	3.93
Course of the blood vessels	3.29	2.86	5.14	5.85	4.14	3.57
Consistency of the blood vessels	4.07	3.21	5.07	5.69	4.93	4.29
Mobility of joints	5.79	6.07	5.93	5.07	5.36	6.50
Mean score (Ø)	3.35	3.67	5.21	4.96	3.99	3.63
Overall mean score (Rats)	4.24					

**Table 3 animals-11-03432-t003:** Anatomic parameters with arithmetic mean values based on the ordinal-scaled variables of the head. Value ranges are as follows: 1–1.49 = very realistic (dark blue; not depicted); 1.5–2.49 = quite realistic (gray blue); 2.5–3.49 = rather realistic (light blue); 3.5–4.49 = rather unrealistic (light orange); 4.5–5.49 = quite unrealistic (red); 5.5–6.5 = very unrealistic (dark red); 7 = not applicable (gray).

HEAD	Simulators					
Simulator	Rat A	Rat B	Rat C	Rat D	Rat E	Mouse
Parameter						
Overall appearance of the head	2.86	3.43	5.43	4.36	3.36	3.43
Head shape	2.29	3.14	5.21	4.50	3.29	3.07
Head proportions	2.36	3.15	5.38	4.50	3.43	3.29
Mobility of the head	3.86	5.50	4.86	3.86	4.00	4.21
Mobility of the lower jaw	4.64	6.07	5.21	7.00	3.29	6.43
Degree of mouth opening	3.50	4.43	4.71	7.00	3.21	5.00
Appearance of the mouth opening	3.57	4.93	5.14	7.00	3.57	5.00
Pharynx and larynx region	3.93	5.36	5.71	7.00	4.07	6.79
Teeth	3.00	7.00	4.79	7.00	3.36	7.00
Tongue	4.36	5.50	4.64	7.00	3.86	7.00
Eyes	4.07	4.29	5.21	4.71	3.86	6.07
Ears	4.93	5.21	5.36	3.29	2.86	4.14
Mean score (Ø)	3.61	4.83	5.14	5.60	3.51	5.12
Overall mean score (Rats)	4.54					

**Table 4 animals-11-03432-t004:** Anatomic parameters with arithmetic mean values based on the ordinal-scaled variables of the tail. Value ranges are as follows: 1–1.49 = very realistic (dark blue; not depicted); 1.5–2.49 = quite realistic (gray blue); 2.5–3.49 = rather realistic (light blue); 3.5–4.49 = rather unrealistic (light orange); 4.5–5.49 = quite unrealistic (red); 5.5–6.5 = very unrealistic (dark red); 7 = not applicable (gray).

TAIL	Simulators					
Simulator	Rat A	Rat B	Rat C	Rat D	Rat E	Mouse
Parameter						
Overall appearance of the tail	2.57	2.93	4.54	5.07	2.50	3.07
Haptic of the tail	2.79	2.93	4.36	4.57	2.43	3.21
Mobility of the tail	2.71	2.64	3.64	3.79	2.36	2.86
Skin texture	2.93	3.36	4.86	4.93	2.71	3.36
Length of the tail	2.00	2.21	4.00	3.29	2.14	2.36
Connection to the trunk/torso	2.92	3.86	5.36	5.93	3.14	3.00
Position of the tail’s blood vessels	2.64	2.29	4.93	5.07	3.50	3.29
Course of the tail’s blood vessels	2.50	2.21	4.86	5.21	3.71	3.00
Consistency of the tail’s blood vessels	3.29	3.00	5.00	5.36	3.93	3.71
Size of the tail’s blood vessels	3.21	2.43	5.14	5.21	3.71	3.21
Visibility of the tail’s blood vessels through the skin	3.64	2.71	4.79	4.93	3.71	3.21
Mean score (Ø)	2.84	2.78	4.68	4.85	3.08	3.12
Overall mean score (Rats)	3.65					

**Table 5 animals-11-03432-t005:** Anatomic parameters with arithmetic mean values based on the ordinal-scaled variables of the limbs. Value ranges are as follows: 1–1.49 = very realistic (dark blue; not depicted); 1.5–2.49 = quite realistic (gray blue); 2.5–3.49 = rather realistic (light blue); 3.5–4.49 = rather unrealistic (light orange); 4.5–5.49 = quite unrealistic (red); 5.5–6.5 = very unrealistic (dark red); 7 = not applicable (gray).

LIMBS	Simulator					
Simulator	Rat A	Rat B	Rat C	Rat D	Rat E	Mouse
Parameter						
Overall appearance of the limbs	3.86	4.36	5.50	4.43	4.07	4.50
Haptic of the limbs	4.79	4.93	5.50	4.43	4.50	5.36
Proportions of the limbs	3.36	3.71	5.14	4.14	3.64	4.29
Posture of the limbs	3.43	4.07	5.64	4.43	4.07	4.21
Mobility of the limbs	4.86	5.07	5.29	4.21	4.79	5.50
Length of the limbs	3.57	3.50	5.14	3.86	3.57	4.14
Toes of the limbs	4.93	5.79	5.29	4.79	3.86	3.79
Mean score (Ø)	4.11	4.49	5.36	4.33	4.07	4.54
Overall mean score	4.47					

**Table 6 animals-11-03432-t006:** Mean rank and overall rank of the rat simulators.

Ranking	Simulators				
	Rat A	Rat B	Rat C	Rat D	Rat E
Mean rank	1.36	2.29	4.43	4.43	2.50
Overall rank	1.	2.	4.	4.	3.

## Data Availability

Data is contained within the article or Appendix A.

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
