# Peer review of "Anatomical Evaluation of Rat and Mouse Simulators for Laboratory Animal Science Courses"

_animals, 2021, doi:10.3390/ani11123432_

Round 1

Reviewer 1 Report

In this study, the authors discussed the importance of applying the 3R principle and described a comparative analysis of rats and mice simulators as a strategy to reduce the number of animals used for education and training. Overall, the study is well written, and the results are well described. I suggest some corrections, based on the comments below:

- It is not clear to what extent the simulators will be relevant for education and training. It seems the use is limited to basic techniques (such as blood collection, gavage, drug administration route), but not at the surgical level. This issue needs to be better addressed to avoid the false impression of the application of the simulators in all sectors of animal experimentation (e.g. surgery). I strongly suggest the addition of a proper description of the simulator's possible applications.

- I suggest adding a table with detailed information about each simulator (e.g. type of material, weight) and perhaps the advantages and limitations for its application in training, aiming to facilitate the reader's understanding of the comparative analysis.

Reviewer 2 Report

Dear authors, 

Congratulations for your work. I think it is a very interesting and methodically well done work. It is a step toward the use of simulators in teaching, and the full replacement of animals in a near future.  

The paper is well written, the data is clearly presented and discussion is well rounded and includes relevant references.

Minor Comments:

1.- I recommend to indicate the colorimetric scale in Table 1, and move it and Table 2, both, to supplementary material. 

2.-I would arrange the legends of the graphs so that they are aligned with the results. "Very relistic" on the right at all and "very unrealistic" on the left, and all along the same lines. However, since you are presenting the tables with the results and in different colours, I would move the graphs to supplementary material. 

3.-  I think it is interesting to show the results of the mouse simulator but no to include in the rank. Thus, e.g.,  in page 16 line 387, the rat simulator B is the second and in the same page line 411, the rat C in the 5th. 

4.- I should include the colorimetric scale in Table 7. 

5.- Descriptive results indicate that rat A is the best model. I do not understand very well the rational to perform an analysis of variance and why including the mouse model. If you want to include this analysis, I should start this section indicating that in the previous study you observed that rat A is the one which obtained the best results so you wanted to have a cuantitative analysis. Then, I would describe the statistical differences results, not including the mouse, and the raw data (tables) can be move to supplementary material. 
